# Hybrid Models for Learning to Branch

**Prateek Gupta**[*]
University of Oxford
The Alan Turing Institute
pgupta@robots.ox.ac.uk

**Maxime Gasse**
Mila, Polytechnique Montréal
maxime.gasse@polymtl.ca

**Elias B. Khalil**
University of Toronto
khalil@mie.utoronto.ca

**M. Pawan Kumar**
University of Oxford
pawan@robots.ox.ac.uk

**Andrea Lodi**
CERC, Polytechnique Montréal
andrea.lodi@polymtl.ca

**Yoshua Bengio**
Mila, Université de Montréal
yoshua.bengio@mila.quebec

## Abstract

A recent Graph Neural Network (GNN) approach for learning to branch has been shown to successfully reduce the running time of branch-and-bound (B&B) algorithms for Mixed Integer Linear Programming (MILP). While the GNN relies on a GPU for inference, MILP solvers are purely CPU-based. This severely limits its application as many practitioners may not have access to high-end GPUs. In this work, we ask two key questions. First, in a more realistic setting where only a CPU is available, is the GNN model still competitive? Second, can we devise an alternate computationally inexpensive model that retains the predictive power of the GNN architecture? We answer the first question in the negative, and address the second question by proposing a new hybrid architecture for efficient branching on CPU machines. The proposed architecture combines the expressive power of GNNs with computationally inexpensive multi-layer perceptrons (MLP) for branching. We evaluate our methods on four classes of MILP problems, and show that they lead to up to 26% reduction in solver running time compared to state-of-the-art methods without a GPU, while extrapolating to harder problems than it was trained on. The code for this project is publicly available at `https://github.com/pg2455/Hybrid-learn2branch`.

## 1 Introduction

Mixed-Integer Linear Programs (MILPs) arise naturally in many decision-making problems such as auction design [1], warehouse planning [13], capital budgeting [14] or scheduling [15]. Apart from a linear objective function and linear constraints, some decision variables of a MILP are required to take integral values, which makes the problem NP-hard [35].

Modern mathematical solvers typically employ the B&B algorithm [29] to solve general MILPs to global optimality. While the worst-case time complexity of B&B is exponential in the size of the problem [38], it has proven efficient in practice, leading to wide adoption in various industries. At a high level, B&B adopts a divide-and-conquer approach that consists in recursively partitioning the original problem into a tree of smaller sub-problems, and solving linear relaxations of the sub-problems until an integral solution is found and proven optimal.

---

[*]The work was done during an internship at Mila and CERC. Correspondence to: <pgupta@robots.ox.ac.uk>

Despite its apparent simplicity, there are many practical aspects that must be considered for B&B to perform well [2]; such decisions will affect the search tree, and ultimately the overall running time. These include several decision problems [33] that arise during the execution of the algorithm, such as *node selection*: which sub-problem do we analyze next?; and *variable selection* (a.k.a. branching): which decision variable must be used (branched on) to partition the current sub-problem? While such decisions are typically made using hard-coded expert heuristics which are implemented in modern solvers, more and more attention is given to statistical learning approaches for replacing and improving upon those heuristics [23; 5; 26; 17; 39]. An extensive review of different approaches at the intersection of statistical learning and combinatorial optimization is given in Bengio et al. [7].

Recently, Gasse et al. [17] proposed to tackle the variable selection problem in B&B using a Graph Neural Network (GNN) model. The GNN exploits the bipartite graph formulation of MILPs together with a shared parametric representation, thus allowing it to model problems of arbitrary size. Using imitation learning, the model is trained to approximate a very good but computationally expensive "expert" heuristic named *strong branching* [6]. The resulting branching strategy is shown to improve upon previously proposed approaches for branching on several MILP problem benchmarks, and is competitive with state-of-the-art B&B solvers. We note that one limitation of this approach, with respect to general B&B heuristics, is that the resulting strategy is only tailored to the class of MILP problems it is trained on. This is very reasonable in our view, as practitioners usually only care about solving very specific problem types at any time.

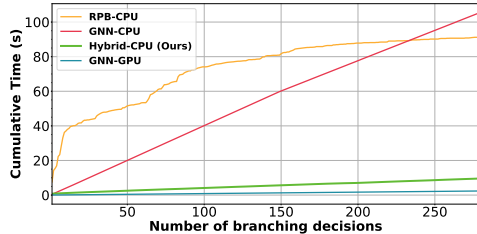

Figure 1: Cumulative time cost of different branching policies: (i) the default internal rule RPB of the SCIP solver; (ii) a GNN model (using a GPU or a CPU); and (iii) our hybrid model. Clearly the GNN model requires a GPU for being competitive, while our hybrid model does not. (Measured on a capacitated facility location problem, medium size).

While the GNN model seems particularly suited for learning branching strategies, one drawback is a high computational cost for inference, i.e., choosing the branching variable at each node of the B&B tree. In Gasse et al. [17], the authors use a high-end GPU card to speed-up the GNN inference time, which is a common practice in deep learning but is somewhat unrealistic for MILP practitioners. Indeed, commercial MILP solvers rely solely on CPUs for computation, and the GNN model from Gasse et al. [17] is not competitive on CPU-only machines, as illustrated in Figure 1. There is indeed a trade-off between the quality of the branching decisions made and the time spent obtaining those decisions. This trade-off is well-known in MILP community [2], and has given rise to carefully balanced strategies designed by MILP experts, such as *hybrid branching* [3], which derive from the computationally expensive *strong branching* heuristic [6].

In this paper, we study the time-accuracy trade-off in learning to branch with the aim of devising a model that is both computationally inexpensive and accurate for branching. To this end, we propose a hybrid architecture that uses a GNN model only at the root node of the B&B tree and a weak but fast predictor, such as a simple Multi-Layer Perceptron (MLP), at the remaining nodes. In doing so, the weak model is enhanced by high-level structural information extracted at the root node by the GNN model. In addition to this new hybrid architecture, we experiment and evaluate the impact of several variants to our training protocol for learning to branch, including: (i) end-to-end training [19; 8], (ii) knowledge distillation [25], (iii) auxiliary tasks [30], and (iv) depth-dependent weighting of the training loss for learning to branch, an idea originally proposed by He et al. [23] in the context of node selection.

We evaluate our approach on large-scale MILP instances from four problem families: Capacitated Facility Location, Combinatorial Auctions, Set Covering, and Independent Set. We demonstrate empirically that our combination of hybrid architecture and training protocol results in state-of-the-art performance in the realistic setting of a CPU-restricted machine. While we observe a slight decrease in the predictive performance of our model with respect to the original GNN from [17], its reduced computational cost still allows for a reduction of up to 26% in overall solving time on all the evaluated benchmarks compared to the default branching strategy of the modern open-source solver SCIP [20]. Also, our hybrid model preserves the ability to extrapolate to harder problems than trained on, as the original GNN model.

## 2  Related Work

Finding a branching strategy that results in the smallest B&B tree for a MILP is at least as hard–and possibly much harder–than solving the MILP with any strategy. Still, small trees can be obtained by using a computationally expensive heuristic named *strong branching* (SB) [6; 31]. The majority of the research efforts in variable selection are thus aimed at matching the performance of SB through faster approximations, via cleverly handcrafted heuristics such as *reliability pseudocost branching* [4], and recently via machine learning[2] [5; 26; 17]. We refer to [33] for an extensive survey of the topic.

Alvarez et al. [5] and Khalil et al. [26] showed that a fast discriminative classifier such as extremely randomized trees [18] or support vector machines [24] on hand-designed features can be used to mimic SB decisions. Subsequently, Gasse et al. [17] and Zarpellon et al. [39] showed the importance of representation learning for branching. Our approach, in some sense, combines the superior representation framework of Gasse et al. [17] with the computationally cheaper framework of Khalil et al. [26]. Such hybrid architectures have been successfully used in ML problems such as visual reasoning [36], style-transfer [11], natural language processing [37; 10], and speech recognition [27].

## 3  Preliminaries

Throughout this paper, we use boldface for vectors and matrices. A MILP is a mathematical optimization problem that combines a linear objective function, a set of linear constraints, and a mix of continuous and integral decision variables. It can be written as:

$$\arg\min_{\mathbf{x}} \mathbf{c}^\mathsf{T}\mathbf{x}, \quad \text{s.t.} \quad \mathbf{A}\mathbf{x} \le \mathbf{b}, \quad \mathbf{x} \in \mathbb{Z}^p \times \mathbb{R}^{n-p},$$

where $\mathbf{c} \in \mathbb{R}^n$ denotes the cost vector, $\mathbf{A} \in \mathbb{R}^{m \times n}$ the matrix of constraint coefficients, $\mathbf{b} \in \mathbb{R}^m$ is the vector of constant terms of the constraints, and there are $p$ integer variables, $1 \le p \le n$ .

The B&B algorithm can be described as follows. One first solves the linear program (LP) relaxation of the MILP, obtained by disregarding the integrality constraints on the decision variables. If the LP solution $\mathbf{x}^\star$ satisfies the MILP integrality constraints, or is worse than a known integral solution, then there is no need to proceed further. If not, then one divides the MILP into two sub-MILPs. This is typically done by picking an integral decision variable that has a fractional value, $i \in \mathcal{C} = \{i \mid x_i^\star \notin \mathbb{Z}, i \le p\}$, and create two sub-MILPs with additional constraints $x_i \le \lfloor x_i^\star \rfloor$ and $x_i \ge \lceil x_i^\star \rceil$, respectively. The decision variable $i$ that is used to partition the feasible region is called the *branching variable*, while $\mathcal{C}$ denotes the *branching candidates*. The second step is to select one of the leaves of the tree, and repeat the above steps until all leaves have been processed[3].

In this work, we refer to the first node processed by B&B as the *root node*, which contains the original MILP, and all subsequent nodes containing a local MILP as *tree nodes*, whenever the distinction is required. Otherwise we refer to them simply as nodes.

## 4  Methodology

As mentioned earlier, computationally heavy GNNs can be prohibitively slow when used for branching on CPU-only machines. In this section we describe our hybrid alternative, which combines the superior inductive bias of a GNN at the root node with a computationally inexpensive model at the tree nodes. We also discuss various enhancements to the training protocol, in order to enhance the performance of the learned models.

### 4.1  Hybrid architecture

A variable selection strategy in B&B can be seen as a scoring function $f$ that outputs a score $s_i \in \mathbb{R}$ for every branching candidate. As such, $f$ can be modeled as a parametric function, learned by ML. Branching then simply involves selecting the highest-scoring candidate according to $f$:

$$i_f^\star = \arg\max_{i \in \mathcal{C}} \mathbf{s}_i$$

Table 1: Various functional forms $f$ considered for variable selection ($\odot$ denotes Hadamard product).

| | Data extraction | Computational Cost | Decision Function |
|---|---|---|---|
| GNN [17] | expensive | Expensive | $\mathbf{s} = \text{GNN}(\mathbf{G})$ |
| MLP | cheap | Moderate | $\mathbf{s} = \text{MLP}(\mathbf{X})$ |
| CONCAT | hybrid | Moderate | $\boldsymbol{\Psi} = \text{GNN}(\mathbf{G}^0)$ <br> $\mathbf{s} = \text{MLP}([\boldsymbol{\Psi}, \mathbf{X}])$ |
| FiLM [36] | hybrid | Moderate | $\boldsymbol{\gamma}, \boldsymbol{\beta} = \text{GNN}(\mathbf{G}^0)$ <br> $\mathbf{s} = \text{FiLM}(\boldsymbol{\gamma}, \boldsymbol{\beta}, \text{MLP}(\mathbf{X}))$ |
| HyperSVM | hybrid | Cheapest | $\mathbf{W} = \text{GNN}(\mathbf{G}^0)$ <br> $\mathbf{s} = (\mathbf{W} \odot \mathbf{X})\mathbf{1}$ |
| HyperSVM-FiLM | hybrid | Cheapest | $\boldsymbol{\gamma}, \boldsymbol{\beta_1}, \boldsymbol{\beta_2} = \text{GNN}(\mathbf{G}^0)$ <br> $\mathbf{s} = \boldsymbol{\beta_2}^\mathsf{T} \max(0, \boldsymbol{\beta_1} \odot \mathbf{X} + \boldsymbol{\gamma})$ |

We consider two forms of node representations for machine learning models: (i) a graph representation $\mathbf{G} \in \mathcal{G}$, such as the variable-constraint bipartite graph of Gasse et al. [17], where $\mathbf{G} = (\mathbf{V}, \mathbf{E}, \mathbf{C})$, with $\mathbf{V} \in \mathbb{R}^{n \times d_1}$ variable features, $\mathbf{E} \in \mathbb{R}^{n \times m \times d_2}$ edge features, and $\mathbf{C} \in \mathbb{R}^{m \times d_3}$ constraint features; and (ii) branching candidate features $\mathbf{X} \in \mathbb{R}^{|\mathcal{C}| \times d_4}$, such as those from Khalil et al. [26], which is cheaper to extract than the first representation. For convenience, we denote by $\mathcal{X}$ the generic space of the branching candidate features, and by $\mathbf{G}^0$ the graph representation of the root node. The various features $d_i$ used in this work are detailed in the supplementary materials.

In B&B, structural information in the tree nodes, $\mathbf{G}$, shares a lot of similarity with that of the root node, $\mathbf{G}^0$. Extracting, but also processing that information at every node is an expensive task, which we will try to circumvent. The main idea of our hybrid approach is then to succinctly extract the relevant structural information only once, at the root node, with a parametric model $\text{GNN}(\mathbf{G}^0; \boldsymbol{\theta})$. We then combine in the tree nodes this preprocessed structural information with the cheap candidate features $\mathbf{X}$, using a hybrid model $f := \text{MLP}_{\text{HYBRID}}(\text{GNN}(\mathbf{G}^0; \boldsymbol{\theta}), \mathbf{X}; \boldsymbol{\phi})$. By doing so, we hope that the resulting model will approach the performance of an expensive but powerful $f := \text{GNN}(\mathbf{G})$, at almost the same cost as an inexpensive but less powerful $f := \text{MLP}(\mathbf{X})$. Figure 2 illustrates the differences between those approaches, in terms of data extraction.

For an exhaustive coverage of the computational spectrum of hybrid models, we consider four ways to enrich the feature space of an MLP via a GNN's output, sumarrized in Table 1. In CONCAT, we concatenate the candidate's *root representations* $\boldsymbol{\Psi}$ with the features $\mathbf{X}$ at a node. In FiLM [36], we generate film parameters $\boldsymbol{\gamma}$, $\boldsymbol{\beta}$ from the GNN, for each candidate, which are further used to modulate the hidden layers of the MLP. In details, if $\boldsymbol{h}$ is the intermediate representation of the MLP, it gets linearly modulated as $\boldsymbol{h} \leftarrow \boldsymbol{\beta} \cdot \boldsymbol{h} + \boldsymbol{\gamma}$. While both the above architectures have similar computational complexity, it has been shown that FiLM subsumes the CONCAT architecture [12]. On the other end of the spectrum lie the most inexpen-

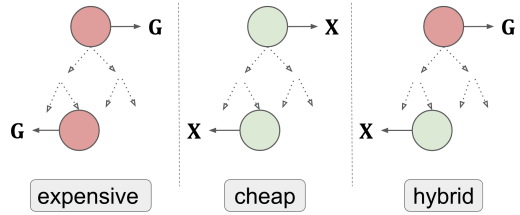

Figure 2: Data extraction strategies: bipartite graph representation $\mathbf{G}$ at every node (expensive); candidate variable features $\mathbf{X}$ at every node (cheap); bipartite graph at the root node and variable features at tree node (hybrid).

sive hybrid architectures, HyperSVM and HyperSVM-FiLM. HyperSVM is inspired by HyperNetworks [22], and simply consists in a multi-class Support Vector Machine (SVM), whose parameters are predicted by the root GNN. We chose a simple linear discriminator, for a minimal computational cost. Finally, in HyperSVM-FiLM we increase the expressivity of HyperSVM with the help of modulations, similar to that of FiLM.

## 4.2 Training Protocol

We use *strong branching* decisions as ground-truth labels for imitation learning, and collect observations of the form $(\mathbf{G}^0, \mathbf{G}, \mathbf{X}, i^\star_{SB})$. Thus, the data used for training the model is $\mathcal{D} =$

$\{(\mathbf{G}_k^0, \mathbf{G}_k, \mathbf{X}_k, i_{SB,k}^\star), k = 1, 2, ..., N\}$. We treat the problem of identifying $i_{SB,k}^\star$ as a classification problem, such that $i^* = \arg\max_{i \in \mathcal{C}} f(\mathbf{G}^0, \mathbf{X})$ is the target outcome. Considering $\mathcal{D}$ as our ground-truth, our objective (1) is to minimize the cross-entropy loss $l$ between $f(\mathbf{G}^0, \mathbf{X}) \in \mathcal{R}^{|\mathcal{C}|}$ and a one-hot vector with one at the target:

$$\mathcal{L}(\mathcal{D}; \boldsymbol{\theta}, \boldsymbol{\phi}) = \frac{1}{N} \sum_{k=1}^N l(f(\mathbf{G}_k^0, \mathbf{X}_k; \boldsymbol{\theta}, \boldsymbol{\phi}), i_{SB,k}^\star). \tag{1}$$

Performance on unseen instances, or generalization, is of utmost importance when the trained models are used on bigger instances. The choices in training the aforementioned architectures influence this ability. In this section, we discuss four such important choices that lead to better generalization.

### 4.2.1 End-to-end Training (e2e)

A GNN$_{\hat{\boldsymbol{\theta}}}$, with pre-trained parameters $\hat{\boldsymbol{\theta}}$, is obtained using the procedure described in Gasse et al. [17]. We use this pre-trained GNN to extract variable representations at the root node and use it as an input to the MLPs at a tree node (pre). This results in a considerable performance boost over plain "salt-of-the-earth" MLP used at all tree nodes. However, going a step further, an end-to-end (e2e) approach involves training the GNN and the MLP together by backpropagating the gradients from a tree node to the root node. In doing so, e2e training aligns the variable representations at the root node with the prediction task at a tree node. At the same time, it is not obvious that it should result in a stable learning behavior because the parameters for GNN need to adapt to various tree nodes. Our experiments explore both pre-training (pre) and end-to-end training (e2e), namely:

$$\text{(pre)} \quad \boldsymbol{\phi}^* = \arg\min_{\boldsymbol{\phi}} \mathcal{L}(\mathcal{D}; \hat{\boldsymbol{\theta}}, \boldsymbol{\phi}), \quad \text{(e2e)} \quad \boldsymbol{\phi}^*, \boldsymbol{\theta}^* = \arg\min_{\boldsymbol{\theta}, \boldsymbol{\phi}} \mathcal{L}(\mathcal{D}; \boldsymbol{\theta}, \boldsymbol{\phi}). \tag{2}$$

### 4.2.2 Knowledge Distillation (KD)

Using the outputs of a pre-trained expert model as a soft-target for training a smaller model has been successfully used in model compression [25]. In this way, one aims to learn an inexpensive model that has the same generalization power as an expert. Thus, for better generalization, instead of training our hybrid architectures with cross-entropy [21] on ground-truth hard-labels, we study the effect of training with KL Divergence [28] between the outputs of a pre-trained GNN and a hybrid model, namely:

$$\text{(KD)} \quad \mathcal{L}_{KD}(\mathcal{D}; \boldsymbol{\theta}, \boldsymbol{\phi}) = \frac{1}{N} \sum_{k=1}^N \text{KL}(f(\mathbf{G}_k^0, \mathbf{X}_k; \boldsymbol{\theta}, \boldsymbol{\phi}), \text{GNN}_{\hat{\boldsymbol{\theta}}}(\mathbf{G_k})). \tag{3}$$

### 4.2.3 Auxiliary Tasks (AT)

An inductive bias, such as GNN, encodes a prior on the way to process raw input. Auxiliary tasks, on the other hand, inject priors in the model through additional learning objectives, which are not directly linked to the main task. These tasks are neither related to the final output nor do they require additional training data. One such auxiliary task is to maximize the diversity in variable representations. The intuition is that very similar representations lead to very close MLP score predictions, which is not useful for branching.

We minimize a pairwise loss function that ensures maximum separation between the variable representations projected on a unit hypersphere. We consider two types of objectives for this: (i) Euclidean Distance (ED), and (ii) Minimum Hyperspherical Energy (MHE) [32], inspired from the well-known Thomson problem [16] in Physics. While ED separates the representations in the Euclidean space on the hypersphere, MHE ensures uniform distribution over the hypersphere. Denoting $\hat{\boldsymbol{\psi}}_i$ as the variable representation for the variable $i$ projected on a unit hypersphere and $e_{ij} = ||\hat{\boldsymbol{\psi}}_i - \hat{\boldsymbol{\psi}}_j||_2$ as the Euclidean distance between the representations for the variables $i$ and $j$, our new objective function is given as $\mathcal{L}_{AT}(\mathcal{D}; \boldsymbol{\theta}, \boldsymbol{\phi}) = \mathcal{L}(\mathcal{D}; \boldsymbol{\theta}, \boldsymbol{\phi}) + g(\boldsymbol{\Psi}; \boldsymbol{\theta})$, where

$$\text{(ED)} \quad g(\boldsymbol{\Psi}; \boldsymbol{\theta}) = \frac{1}{N^2} \sum_{i,j=1}^N e_{ij}^2, \quad \text{(MHE)} \quad g(\boldsymbol{\Psi}; \boldsymbol{\theta}) = \frac{1}{N^2} \sum_{i,j=1}^N \frac{1}{e_{ij}}. \tag{4}$$

#### 4.2.4 Loss Weighting Scheme

The problem of distribution shift is unavoidable in a sequential process like B&B. A suboptimal branching decision at a node closer to the root node can have worse impact on the size of the B&B tree as compared to when such a decision is made farther from it. In such situations, one can use depth (possibly normalized) as a feature, but the generalization on bigger instances is a bit unpredictable as the distribution of this feature might be very different from that observed in the training set. Thus, we experimented with different depth-dependent formulations for weighting the loss at any node. Denoting $z_i$ as the depth of a tree node $i$ relative to the depth of the tree, we weight the loss at different tree nodes by $w(z_i)$, making our objective function as

$$\mathcal{L}(\mathcal{D}; \boldsymbol{\theta}, \boldsymbol{\phi}) = \frac{1}{N} \sum_{k=1}^{N} w(z_k) \cdot l(f(\mathbf{G}_k^0, \mathbf{X}_k; \boldsymbol{\theta}, \boldsymbol{\phi}), i^\star_{SB,k}). \tag{5}$$

Specifically, we considered 5 different weighting functions such that all of them have the same end points, i.e., $w(0) = 1.0$ at the root node and $w(1) = e^{-0.5}$ at the deepest node. Different functions were chosen depending on their intermediate behaviour in between these two points. We experimented with exponential, linear, quadratic and sigmoidal decay behavior of these functions. Table 3 lists various functions and their mathematical forms considered in our experiments.

## 5 Experiments

We follow the experimental setup of Gasse et al. [17], and evaluate each branching strategy across four different problem classes, namely Capacitated Facility Location, Minimum Set Covering, Combinatorial Auctions, and Maximum Independent Set. Randomly generated instances are solved offline using SCIP [20] to collect training samples of the form $(\mathbf{G}^0, \mathbf{G}, \mathbf{X}, i^\star_{SB})$. We leave the description of the data collection and training details of each model to the supplementary materials.

**Evaluation.** As in Gasse et al. [17], our evaluation instances are labeled as small, medium, and big based on the size of underlying MILP. Small instances have the same size as those used to generate the training datasets, and thus match the training distribution, while instances of increasing size allows us to measure the generalization ability of the trained models. Each scenario uses 20 instances, solved using 3 different seeds to account for solver variability. We report standard metrics used in the MILP community for benchmarking B&B solvers: (i) Time: 1-shifted geometric mean[4] of running times in seconds, including the running times for unsolved instances, (ii) Nodes: hardware-independent 1-shifted geometric mean of B&B node count of the instances solved by all branching strategies , and (iii) Wins: number of times each branching strategy resulted in the fastest solving time, over total number of solved instances. All branching strategies are evaluated using the open-source solver SCIP [20] with a time limit of 45 minutes, and cutting planes are allowed only at the root node.

**Baselines.** We compare our hybrid model to several non-ML baselines, including SCIP's default branching heuristic *Reliability Pseudocost Branching* (RPB), the "gold standard" heuristic *Full Strong Branching* (FSB), and a very fast but ineffective heuristic *Pseudocost Branching* (PB). We also include the GNN model from Gasse et al. [17] run on CPU (GNN), and also several fast but less expressive models such as SVMRank from Khalil et al. [26], LambdaMART from

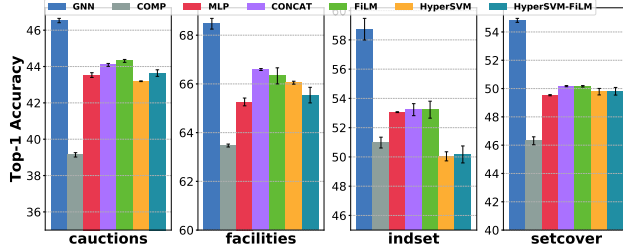

Figure 3: Test accuracy of the different models, with a simple e2e training protocol.

Burges [9], and ExtraTree Classifier from Geurts et al. [18] as benchmarks. For conciseness, we chose to report those last three competitor models using an optimistic aggregation scheme, by systematically choosing only the best performing method among the three (COMP). For completeness, we also

report the performance of a GNN model run on a high-end GPU, although we do not consider that method as a baseline and therefore do not include it in the Wins indicator. We acknowledge that our comparison to general branching strategy like RPB is not completely fair, however, developing specialized versions of such strategies is a challenge in itself full of modeling choices that we foresee as future work.

**Model selection.** To investigate the effectiveness of different architectures, we empirically compare the performance of their end-to-end variants. Figure 3 compares the Top-1 test accuracy of models across the four problem sets. The performance of GNNs (blue), being the most expressive model, serves as an upper bound to the performance of hybrid models. All of the considered hybrid models outperform MLPs (red) across all problem sets. Additionally, we observe that FiLM (green) and CONCAT (purple) perform significantly better than other architectures. However, there is no clear winner among them. We also note that the cheapest hybrid models, HyperSVM and HyperSVM-FiLM, though better than MLPs, still do not perform as well as FiLM or CONCAT models.

**Training protocols.** In Table 2, we show the effect of different protocols discussed in section 4 on Top-1 accuracy of the FiLM models. We observe that the presence of these protocols improves the model accuracy by 0.5-0.9%, which translates to a minor yet practically useful improvement in B&B performance of the solver. Except for Combinatorial Auctions, FiLM's performance is improved by knowledge distillation, which suggests that the soft targets of pre-trained GNN yield a better generalization performance. Lastly, we launch a hyperparameter search for auxiliary objective: ED and MHE, on top of the best performing model (Top-1 accuracy), among e2e and e2e & KD models. Auxiliary tasks further help the accuracy of the hybrid models, but for some problem classes it is ED that works well while for others it is MHE. We provide the model performances on test dataset in the supplement for further reference.

**Effect of loss weighting.** We empirically investigate the effect of the different loss weighting schemes discussed in Section 4.2.4. We train a simple MLP model on our small Combinatorial Auctions instances, and measure the resulting B&B tree size on big instances. We report aggregated results in Table 3, and provide instance-level results in the supplement. We observe that the most commonly used exponential and linear schemes actually seem to degrade the performance of the learned strategy, as we believe those may be too aggressive at disregarding nodes early on in the tree.

Table 2: Test accuracy of FiLM, using different training protocols.

|  | cauctions | facilities | indset | setcover |
|---|---|---|---|---|
| Pretrained GNN | $44.12 \pm 0.09$ | $65.78 \pm 0.06$ | $53.16 \pm 0.51$ | $50.00 \pm 0.09$ |
| e2e | $44.31 \pm 0.08$ | $66.33 \pm 0.33$ | $53.23 \pm 0.58$ | $50.16 \pm 0.05$ |
| e2e & KD | $44.10 \pm 0.09$ | $66.60 \pm 0.21$ | $53.08 \pm 0.3$ | $50.31 \pm 0.19$ |
| e2e & KD & AT | $\mathbf{44.56 \pm 0.13}$ | $\mathbf{66.85 \pm 0.28}$ | $\mathbf{53.68 \pm 0.23}$ | $\mathbf{50.37 \pm 0.03}$ |

Table 3: Effect of different sample weighting schemes on combinatorial auctions (big) instances, with a simple MLP model. $z \in [0, 1]$ is the ratio of the depth of the node and the maximum depth observed in a tree.

| Type | Weighting scheme | Nodes | Wins |
|---|---|---|---|
| Constant | $1$ | 9678 | 10/60 |
| Exponential decay | $e^{-0.5z}$ | 9793 | 10/60 |
| Linear | $(e^{-0.5} - 1) * z + 1$ | 9789 | 12/60 |
| Quadratic decay | $(e^{-0.5} - 1) * z^2 + 1$ | 9561 | 14/60 |
| Sigmoidal | $(1 + e^{-0.5})/(1 + e^{z-0.5})$ | **9534** | 14/60 |

On the other hand, both the quadratic and sigmoidal schemes result in an improvement, thus validating the idea that depth-dependent weighting can be beneficial for learning to branch. We therefore opt for a sigmoidal loss weighting scheme in our training protocol.

**Complete benchmark.** Finally, to evaluate the runtime performance of our hybrid approach, we replace SCIP's default branching strategy with our best performing model, FiLM. We observe in Table 4 that FiLM performs substantially better than all other CPU-based branching strategies. The computationally expensive FSB, our "gold standard", becomes impractical as the size of instances grows, whereas RPB remains competitive. While GNN retains its small number of nodes, it loses in running time performance on CPU. Note that we found that FiLM models for Maximum Independent Set did initially overfit on small instances, such that the performance on larger instances degraded substantially. To overcome this issue we used weight decay [34] regularization, with a validation set of 2000 observations generated using random medium instances (not used for evaluation). We report the performance of the regularized models in the supplement, and use the best performing model to report evaluation performance on medium and big instances. We also show in the supplement that the

Table 4: Performance of *branching strategies* on evaluation instances. We report geometric mean of solving times, number of times a method won (in solving time) over total finished runs, and geometric mean of number of nodes. Refer to section 5 for more details. The best performing results are in **bold**. *Models were regularized to prevent overfitting on small instances.

| Model | Small | | | Medium | | | Big | | |
|---|---|---|---|---|---|---|---|---|---|
| | Time | Wins | Nodes | Time | Wins | Nodes | Time | Wins | Nodes |
| FSB | 42.53 | 1 / 60 | 13 | 313.33 | 0 / 59 | 75 | 997.23 | 0 / 51 | 50 |
| PB | 31.35 | 4 / 60 | 139 | 177.69 | 4 / 60 | 384 | 712.45 | 3 / 56 | 309 |
| RPB | 36.86 | 1 / 60 | **23** | 213.99 | 1 / 60 | **152** | 794.80 | 2 / 54 | **99** |
| COMP | 30.37 | 3 / 60 | 120 | 172.51 | 4 / 60 | 347 | 633.42 | 6 / 57 | 294 |
| GNN | 39.18 | 0 / 60 | 112 | 209.84 | 0 / 60 | 314 | 748.85 | 0 / 54 | 286 |
| FiLM (ours) | **24.67** | **51** / 60 | 109 | **136.42** | **51** / 60 | 325 | **531.70** | **46** / 57 | 295 |
| GNN-GPU | 28.91 | – / 60 | 112 | 150.11 | – / 60 | 314 | 628.12 | – / 56 | 286 |
| | | | | Capacitated Facility Location | | | | | |
| FSB | 27.16 | 0 / 60 | 17 | 582.18 | 0 / 45 | 116 | 2700.00 | 0 / 0 | n/a |
| PB | 10.19 | 0 / 60 | 286 | 94.12 | 0 / 60 | 2451 | 2208.57 | 0 / 23 | 82 624 |
| RPB | 14.05 | 0 / 60 | **54** | 94.65 | 0 / 60 | 1129 | 1887.70 | 7 / 27 | 48 395 |
| COMP | 9.83 | 3 / 60 | 178 | 89.24 | 0 / 60 | 1474 | 2166.44 | 0 / 21 | 52 326 |
| GNN | 17.61 | 0 / 60 | 136 | 242.15 | 0 / 60 | **1013** | 2700.17 | 0 / 0 | n/a |
| FiLM (ours) | **8.73** | **57** / 60 | 147 | **63.75** | **60** / 60 | 1131 | **1843.24** | **20** / 26 | **37 777** |
| GNN-GPU | 8.26 | – / 60 | 136 | 53.56 | – / 60 | 1013 | 1535.80 | – / 36 | 31 662 |
| | | | | Set Covering | | | | | |
| FSB | 6.12 | 0 / 60 | 6 | 132.38 | 0 / 60 | 71 | 2127.35 | 0 / 28 | 318 |
| PB | 2.76 | 1 / 60 | 234 | 25.83 | 0 / 60 | 2765 | 393.60 | 0 / 59 | 13 719 |
| RPB | 4.01 | 0 / 60 | **11** | 26.36 | 0 / 60 | 714 | **210.95** | **29** / 60 | 4701 |
| COMP | 2.76 | 0 / 60 | 82 | 29.76 | 0 / 60 | 930 | 494.59 | 0 / 54 | 5613 |
| GNN | 2.73 | 1 / 60 | 71 | 22.26 | 0 / 60 | 688 | 257.99 | 6 / 60 | **3755** |
| FiLM (ours) | **2.13** | **58** / 60 | 73 | **15.71** | **60** / 60 | **686** | 217.02 | 25 / 60 | 4315 |
| GNN-GPU | 1.96 | – / 60 | 71 | 11.70 | – / 60 | 688 | 121.18 | – / 60 | 3755 |
| | | | | Combinatorial Auctions | | | | | |
| FSB | 673.43 | 0 / 53 | 47 | 1689.75 | 0 / 20 | 10 | 2700.00 | 0 / 0 | n/a |
| PB | 172.03 | 2 / 57 | 5728 | 753.95 | 0 / 45 | 1570 | 2685.23 | 0 / 1 | 38 215 |
| RPB | 59.87 | 5 / 60 | 603 | 173.17 | 11 / 60 | **205** | 1946.51 | 9 / 21 | 2461 |
| COMP | 82.22 | 1 / 58 | 847 | 383.97 | 1 / 52 | 267 | 2393.75 | 0 / 6 | 5589 |
| GNN* | **44.07** | 15 / 60 | **331** | 625.23 | 1 / 50 | 599 | 2330.95 | 0 / 10 | **687** |
| FiLM* (ours) | 52.96 | 37 / 55 | 376 | **131.45** | 47 / 54 | 264 | **1823.29** | 12 / 15 | 1201 |
| GNN-GPU* | 31.71 | – / 60 | 331 | 63.96 | – / 60 | 599 | 1158.59 | – / 27 | 685 |
| | | | | Maximum Independent Set | | | | | |

cheapest computation model of HyperSVM/HyperSVM-FiLM do not provide any runtime advantages over FiLM. We achieve up to 26% reduction on medium instances and up to 8% reduction on big instances in overall solver running time compared to the next-best branching strategy, including both learned and classical strategies.

Finally, we note that the majority of "big" problems in Set Covering and Maximum Independent Set are not solved by any of the branching strategies. Therefore, we provide a comparison of optimality gap of unsolved instances at time out in Table 5. It is evident that FiLM models are able to close a larger optimality gap than the other branching strategies.

In the absence of significant difference in KD and KD & AT model's Top-1 accuracy (see Table 2), a natural question then to ask is: what can be a practically useful choice? To answer this question, we tested the effect of each training protocol on the final B&B performance. Although the detailed discussion is in the supplement we conclude that in the absence of significant difference in the test accuracy of models between KD and AT & KD, a preference should be made for KD models to attain better generalization.

Table 5: Mean optimality gap (lower the better) of commonly unsolved "big" instances (number of such instances in brackets).

| | setcover (33) | indset (39) |
|---|---|---|
| FSB | 0.1709 | 0.0755 |
| PB | 0.0713 | 0.0298 |
| RPB | 0.0628 | 0.0252 |
| COMP | 0.0740 | 0.0252 |
| GNN | 0.1039 | 0.0341 |
| FiLM | **0.0597** | **0.0187** |

**Limitations.** We would like to point out some limitations of our work. First, given the NP-Hard nature of MILP solving, it is fairly time consuming to evaluate performance of the trained models on the instances bigger than considered for this work. One can consider the primal-dual bound gap after a time limit as an evaluation metric for the bigger instances, but this is misaligned with the solving

time objective. Second, we have used Top-1 accuracy on the test set as a proxy for the number of nodes, but there is an inherent distribution shift because of the sequential nature of B&B that leads to out-of-distribution observations. Third, generalization to larger instances is a central problem in the design of branching strategies. Several techniques discussed in this work form only a part of the solution to this problem. On the Maximum Independent Set problem, we originally noticed a poor generalization capability, which we addressed by cross-validation using a small validation set. In future work, we plan to perform an extensive study of the effect of architecture and training protocols on generalization performance. Another experiment worth conducting would be to train on larger instances than "small" problems used in the work, in order to get a better view of how and to which point the different models are able to generalize. However, not only is this a time-consuming process, but there is also an upper limit on the size of the problems on which it is reasonable to conduct experiments, simply due to hardware constraints. Finally, although we showed the efficacy of our models on a broad class of MILPs, there may be other problem classes for which our models might not result in a substantial runtime improvements.

# 6   Conclusion

As more operations research and integer programming tools start to include ML and deep learning modules, it is necessary to be mindful of the practical bottlenecks faced in that domain. To this end, we combine the expressive power of GNNs with the computational advantages of MLPs to yield novel hybrid models for learning to branch, a central component in MILP solving. We integrate various training protocols that augment the basic MLPs and help bridge the accuracy gap with more expensive models. This competitive accuracy translates into savings in time and nodes when used in MILP solving, as compared to both default, expert-designed branching strategies and expensive GNN models, thus obtaining the "best of both worlds" in terms of the time-accuracy trade-off. More broadly, our philosophy revolves around understanding the intricacies and practical constraints of MILP solving and carefully adapting deep learning techniques therein. We believe that this integrative approach is crucial to the adoption of statistical learning in exact optimization solvers.

## Broader Impact

This paper establishes a bridge between the work done in ML in the last years on learning to branch and the traditional MILP solvers used to routinely solve thousands of optimization applications in energy, telecommunications, logistics, biology, just to mention a few.

The MILP solvers are executed on CPU-only machines and the technical challenge of using GPU-based algorithmic techniques to hybridize them had been neglected thus far. Admittedly, such a challenge was not urgent when the learning to branch literature was in its early stages. That situation has changed drastically with the GNN implementation in [17], the first approach to show significant benefit with respect to the default version of a state-of-the-art MILP solver like SCIP. For this reason, the current paper comes at the due time for the literature in the field and addresses the challenge, for the first time, in a sophisticated, yet relatively simple way.

Thus, our work provides the first viable way for commercial and noncommercial MILP solver developers to implement and integrate a ML-based "learning to branch" framework and for hundreds of thousands of users and practitioners to use it. In an even broader sense, the fact that we were able to approximate the performance of GPU-based models with a sophisticated integration of CPU-based techniques is consistent with, for example, Hinton et al. [25], and widens the space of problems to which ML techniques can be successfully applied.

## Acknowledgments and Disclosure of Funding

The authors are grateful to CIFAR and IVADO for funding and Compute Canada for computing resources.

We would further like to acknowledge the important role played by our colleagues at Mila and CERC through building a fun learning environment. We would also like to thank Felipe Serano and Benjamin Müller for their technical help with SCIP and insightful discussions on branching in MILPs. PG wants to thank Giulia Zarpellon, Didier Chételat, Antoine Prouvost, Karsten Roth, David

Yu-Tung Hui, Tristan Deleu, Maksym Korablyov, and Alex Lamb for enlightening discussions on deep learning and integer programming.

## Footnotes

[2] Interestingly, early works on ML methods for branching can be traced back to 2000 [2, Acknowledgements].

[3] For a more involved description of B&B, the reader is referred to Achterberg [2].

[4]for complete definition refer to Appendix A.3 in Achterberg [2]

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
