[Supplementary Material]

# Supplement: Hybrid Models for Learning to Branch

**Prateek Gupta**[*]
University of Oxford
The Alan Turing Institute

**Maxime Gasse**
Mila, Polytechnique Montréal

**Elias B. Khalil**
University of Toronto

**M. Pawan Kumar**
University of Oxford

**Andrea Lodi**
CERC, Polytechnique Montréal

**Yoshua Bengio**
Mila, Université de Montréal

## 1   Inefficiency in using GNNs for solving MILPs in parallel

In this section, we argue that the GNN architecture looses its advantages in the face of solving multiple MILPs at the same time. In the applications like multi-objective optimization [4], where multiple MILPs are solved in parallel, a GNN for each MILP needs to be initialized on the GPU because of the *sequentially asynchronous* nature of solving MILPs. Not only is there a limit to the number of such GNNs that can fit on a single GPU because of memory constraints, but also several GNNs on a single GPU results in an inefficient GPU utilization.

One can, for instance, try to time multiple MILPs such that there is a need for a single forward evaluation on a GPU, but, in our knowledge, it has not been done and it results in frequent interruptions in the solving procedure. An alternative, much simpler, method is to *pack* multiple GNNs on a single GPU such that each GNN is dedicated to solving one MILP. For example, we were able to put 25 GNNs on Tesla V100 32 GB GPU. Figure 1 shows the inefficient utilization of GPUs when multiple GNNs are packed on a single GPU.

Figure 1: *Packing* several GNNs together on a GPU keeps it underutilized. "Size" is the number of inputs batched together (unrealistic scenario) or number of inputs simultaneously put on a GPU separately.

---

[*]The work was done during an internship at Mila and CERC. Correspondence to: <pgupta@robots.ox.ac.uk>

## 2 Input Features

We use the features that were used by Gasse et al. [1] and Khalil et al. [3]. The list of features in **G** are described in the Table 1. There are a total of 92 features that we use for **X** as described in the Table 2. We follow the preprocessing procedure as described in Gasse et al. [1].

Table 1: Features of **G**. It is constructed as a variable-constraint bipartite graph, where the vertices of this graph have features as described here. These features are same as used in Gasse et al. [1].

|  | Type | Description |
|---|---|---|
| **Variable Features (Total -13)** | | |
| Variable type (1) | Categorical | Allowed values - Binary, Integer, Continuous, Implied Integer |
| Normalized coefficient (1) | Real | Objective coefficient of the variable normalized by the euclidean norm of its coefficients in the constraints |
| Specified bounds (2) | Binary | Does the variable has a lower bound (upper bound)? |
| Solution bounds (2) | Binary | If the variable is currently at its lower bound (upper bound)? |
| Solution bractionality (1) | Real $\in [0, 1)$ | Fractional part of the variable i.e. $x - \lfloor x \rfloor$, where $x$ is the value of the decision in variable in the current LP solution |
| Basis (1) | Categorical | One of 4 classes - Lower (variable is at the lower bound), Basic (variable has a value between the bounds), Upper (variable is at the upper bound), Zero (rare case) |
| Reduced cost (1) | Real | Amount by which objective coefficient of the variable should decrease so that the variable assumes a positive value in the LP solution |
| Age (1) | Real | Number of LP iterations since the last time the variable was basic normalized by total number of LP iterations |
| Solution value (1) | Real | Value of the variable in the current LP solution |
| Primal value (1) | Real | Value of the variable in the current best primal solution |
| Average primal value (1) | Real | Average value of the variable in all of the previously observed feasible primal solutions |
| **Constraint Features (Total - 5)** | | |
| Cossine similarity (1) | Real | Cossine of angle between the vector represented by objective coefficients and the coefficients of this constraint |
| Bias (1) | Real | Right hand side of the constraint normalized by the euclidean norm of the row coefficients |
| Age (1) | Real | Number of iterations since the last time the constraint was active normalized by total number of LP iterations |
| Normalized dual value (1) | Real | Value of dual variable corresponding to the constraint normalized by the product of norms of the row coefficients and the objective coefficients |
| Bounds (1) | Binary | If the constraint is currently at its bounds? |
| **Edge Features (Total - 1)** | | |
| Normalized coefficient (1) | Real | Coefficient of the variable normalized by the norm of the coefficients of all the variables in the constraint |

Table 2: Features in **X**, an input to MLP.

|  | count |
|---|---|
| Variable features from **G** | 13 |
| Variable features from Khalil et al. [3] | 72 |

# 3 Preliminary results on running times of various architectures

In order to have a rough idea of relative improvement in runtimes across various architectures, we considered 20 instances in each difficulty level for each problem class and use the solver to obtain observations at each node. Different functions are then used to evaluate decisions at each node, and the total time taken for all the function evaluations across the nodes in an instance is observed. Note that the quality of decision is not important here; we are only interested in time taken per decision.

Specifically, we considered 5 forms of architectures as listed in Table 3. To cover the entire spectrum of strong inductive biases, we constructed an attention mechanism as explained in 4. At the same time, to cover the spectrum of cheaper architectures we consider simple dot product as the form of predictor. Finally, we consider a scenario where we only consider MLPs as predictors everywhere in the B&B tree.

Table 3: Different architectures considered for preliminray runtime comparison.

| Type | Description |
|---|---|
| GNN ALL | Use a Graph Convolution Neural Network (GNN) [1] at all *tree nodes* |
| ATTN ALL | Attention mechanism (section 4) at all nodes |
| GNN DOT | Use GNN at the *root nodes* and a dot product with $\mathbf{X}$ to compute scores at every node |
| ATTN DOT | Use attention mechanism at the root node and dot product with $\mathbf{X}$ to compute scores at every node |
| MLP ALL | Use a 3-layer multilinear perceptron at all the nodes |

Figure 2: Relative time performance of various methods with respect to GNN ALL on CPU (average values). A value of 10 implies that the method is 10 times faster (on arithmetic average) than GNN ALL implying that one can afford to perform 10 times worse than GNN ALL in iterative performance when using CPUs. Note that this is just a rough estimation.

Figure 2 shows the relative performance (rel. to GNN) of various deep learning architectures across 4 sets of problems. It is evident that MLP ALL and GNN DOT are favored across the problem sets. This observation inspired the range of *hybrid architectures* that we explored in the paper. We also observe that a superior inductive bias like that of attention mechanism and transformers [7] has a better runtime performance on GPUs as illustrated in Figure 3, which we suspect is massive parallelization employed in the computations of attention. However, their performance on CPUs is not better than GNNs.

Figure 3: Relative time performance of various methods with respect to $GNNALL$ on GPU (average values). A value of $10$ implies that the method is $10$ times faster (on arithmetic average) than $GNNALL$ implying that one can afford to perform $10$ times worse than $GNNALL$ in iterative performance when using CPUs. Note that this is just a rough estimation.

## 4 Attention Mechanism for MILPs

To cover the entire spectrum of computational complexity and expressivity of inductive bias, we implemented a transformer [7] as an architecture to replace GNNs. Specifically, we let the variables (constraints) attend to all other variables (constraints) via multi-headed self-attention mechanism. Finally, a modulated attention mechanism between *variable representations* as queries and *constraint representation* as keys outputs final *variable representations*, which is passed through the softmax layer for classification objective. Here, we use modulation scheme as explained in Shaw et al. [6]. Precisely, an edge in variable-constraint graph is used to increment the attention score with a learnable scalar value.

Figure 4: Multi-Head Attention mechanism where a variable or constraint can attend to all other variables or constraints. Finally,variables are used as a query to attend to constraints, where the attention is modulated through variable-constraint features. Modulation of attention scores follow Shaw et al. [6]

# 5  Data Generation & Training Specifications

For our experiments, we used 10,000 training instances for the training dataset and collected 150,000 observations from the tree nodes of those instances. In a similar manner, we generated 20,000 instances each for the validation and testing set, resulting in 30,000 observations each for validation and testing respectively.

We held all the training parameters fixed across the models. Specifically, we used a learning rate of $1e^{-3}$, training batch size of 32, a learning rate scheduler to reduce learning rate by $0.2$ if there is no improvement in the validation loss for $15$ epochs, and an early stopping criterion of 30 epochs. Our epoch consisted of 10K training examples and 2K validation samples. We used a 3 layered MLP with 256 hidden units in each layer, while we used GNN model with an embedding size of 64 units. We used this configuration across all the models discussed in the main paper. Due to the large size of instances in capacitated facility location, we used a learning rate of 0.005, early stopping criterion of 20 epochs with a patience of 10 epochs.

Further, for knowledge distillation we used $T = 2$ (temperature) and $\alpha = 0.9$ (convex mixing of soft and hard objectives). These are the recommended settings in Hinton et al. [2]. We did a hyperparameter search for $\beta = \{0.01, 0.001, 0.0001\}$ for ED and MHE. Following are the values for $\beta$ that resulted in the best performing models.

Table 4: Best AT models

|  | AT |
| --- | --- |
| Capacitated Facility Location | MHE |
| Combinatorial Auctions | MHE |
| Set Covering | ED |
| Maximum Independent Set | MHE |

We implemented all the models using PyTorch [5], and ran all the CPU evaluations on an Intel(R) Xeon(R) CPU E5-2650 v4 @ 2.20GHz. GPU evaluations for GNN were performed on NVIDIA-TITAN Xp GPU card with CUDA 10.1.

# 6  Depth-dependent loss weighting scheme

In Figure 5 we plot the sorted ratio of number of nodes to the minimum number of nodes observed for an instance across all the weighting schemes. Here we attempt to breakdown the performance of different branching strategies learned using different loss-weighting schemes. We observe that the *sigmoidal* scheme achieves the best performance.

Figure 5: Performance of different weighing schemes across "big" instances. "Sigmoidal" evolves slowest among all.

# 7   Model Results

The table below is a list of all the architectures along with their performance on the test sets. Please note that training with auxiliary task is not possible with HyperSVM type of architecture so their results are not reported in the table.

Table 5: Top-1 accuracy of various models.

|  |  | Combinatorial Auctions | Capacitated Facility Location | Set Cover | Maximum Independent Set |
|---|---|---|---|---|---|
| Expert | GCNN | 46.53 ± 0.12 | 68.47 ± 0.22 | 54.82 ± 0.14 | 58.73 ± 0.54 |
| Existing methods | Extratrees | 37.97 ± 0.1 | 60.06 ± 0.18 | 42.66 ± 0.22 | 19.35 ± 0.55 |
|  | LMART | 38.7 ± 0.16 | 63.28 ± 0.06 | 46.31 ± 0.28 | 50.98 ± 0.37 |
|  | SVMRank | 39.14 ± 0.12 | 63.47 ± 0.06 | 46.23 ± 0.11 | 49.29 ± 0.23 |
| Simple Network | MLP | 43.53 ± 0.13 | 65.26 ± 0.16 | 49.53 ± 0.04 | 53.06 ± 0.03 |
| Concatenate | CONCAT (Pre) | 44.16 ± 0.03 | 65.5 ± 0.1 | 49.98 ± 0.18 | 52.85 +- 0.34 |
|  | CONCAT (e2e) | 44.09 ± 0.08 | 66.59 ± 0.04 | 50.17 ± 0.04 | 53.23 ± 0.41 |
|  | CONCAT (e2e & KD) | 44.19 ± 0.05 | 66.53 ± 0.13 | 50.11 ± 0.09 | 53.66 +- 0.32 |
| Modulate | FiLM (Pre) | 44.12 ± 0.09 | 65.78 ± 0.06 | 50.0 ± 0.09 | 53.16 ± 0.51 |
|  | FiLM (e2e) | 44.31 ± 0.08 | 66.33 ± 0.33 | 50.16 ± 0.05 | 53.23 ± 0.58 |
|  | FiLM (e2e & KD) | 44.1 ± 0.09 | 66.6 ± 0.21 | 50.31 ± 0.19 | 53.08 ± 0.3 |
| HyperSVM | HyperSVM (e2e) | 43.19 ± 0.02 | 66.05 ± 0.06 | 49.78 ± 0.23 | 50.04 ± 0.31 |
|  | HyperSVM (e2e & KD) | 42.55 ± 0.03 | 66.07 ± 0.05 | 49.53 ± 0.13 | 49.34 +- 0.43 |
| HyperSVM-FiLM | HyperSVM-FiLM (e2e) | 43.64 ± 0.18 | 65.54 ± 0.32 | 49.81 ± 0.27 | 50.17 ± 0.58 |
|  | HyperSVM-FiLM (e2e & KD) | 43.28 ± 0.48 | 65.52 ± 0.34 | 49.73 ± 0.05 | 49.73 +- 0.39 |
|  | FiLM (e2e & KD & AT) | **44.56 ± 0.13** | **66.85 ± 0.28** | **50.37 ± 0.03** | **53.68 ± 0.23** |

# 8   Overfitting in maximum independent set

Table 6 shows that the FiLM models overfit on small instances of maximum independent set. To address this problem, we used a mini dataset of 2000 observations obtained by running data collection on medium instances of maximum independent set. Further, we regularized the FiLM parameters of the FiLM model to yield much simpler models based on the performance on this mini-dataset, which is not too expensive to obtain owing to the size of observations. The results of the regularized models are in Table 7 For a fair comparison, we regularized GNN and used the best performing model to report evaluation results in the main paper.

Table 6: FiLM models for maximum independent set overfits on small instances

|  | small | | | medium | | | big | | |
|---|---|---|---|---|---|---|---|---|---|
|  | Time | Wins | Nodes | Time | Wins | Nodes | Time | Wins | Nodes |
| FiLM | 52.96 | **39**/ 55 | 492 | 1515.19 | 9/ 17 | 2804 | 2700.02 | 0/ 0 | nan |
| GNN-CPU | **44.07** | 21/ 60 | **432** | **371.81** | **36**/ 40 | **558** | **1981.43** | **10**/ 10 | **6334** |
| GNN-GPU | 31.70 | 0/ 59 | 432 | 264.01 | 0/ 43 | 558 | 1772.12 | 0/ 13 | 6313 |

Table 7: Top-1 accuracy of regularized models on 2000 observations from medium random instances of maximum independent set.

| weight decay | FiLM | GNN |
|---|---|---|
| 1.0 | 55.15 ± 0.07 | 31.6 ± 6.63 |
| 0.1 | **56.13 ± 0.32** | **37.23 ± 0.92** |
| 0.01 | 53.25 ± 0.95 | 26.8 ± 16.71 |
| 0.0 | 19.18 ± 4.24 | 34.08 ± 4.8 |

# 9 Performance of HyperSVM architectures

HyperSVM architectures are the cheapest in computation. In this section we ask if we are willing to lose machine learning accuracy by 1-2% in HyperSVM architecture, can we still get faster running times with HyperSVM type of architectures? Table 8 compares solver performance by using HyperSVM architecture and FiLM architecture. We observe that HyperSVM types of architectures loose their ability to generalize on larger scale instances.

Table 8: FiLM architecture generalizes to larger instances better than the HyperSVM types of architectures. We compare the performance of the best FiLM architecture from the Table 5 with the best HyperSVM architecture in the same table.

| | small | | | medium | | | big | | |
| | Time | Wins | Nodes | Time | Wins | Nodes | Time | Wins | Nodes |
|---|---|---|---|---|---|---|---|---|---|
| FiLM | **24.67** | **53**/ 60 | **109** | **136.42** | **51**/ 60 | **336** | **531.70** | **46**/ 57 | **345** |
| HyperSVM | 27.26 | 7/ 60 | 110 | 158.97 | 9/ 60 | 345 | 614.36 | 11/ 57 | 346 |
| MLP | 27.61 | 4/ 60 | 114 | 156.30 | 11/ 60 | 347 | 595.31 | 9/ 56 | 334 |

(a) Capacitated Facility Location

| | small | | | medium | | | big | | |
| | Time | Wins | Nodes | Time | Wins | Nodes | Time | Wins | Nodes |
|---|---|---|---|---|---|---|---|---|---|
| FiLM | **8.73** | **59**/ 60 | **147** | **63.75** | **60**/ 60 | **2169** | **1843.24** | **24**/ 26 | **38530** |
| HyperSVM-FiLM | 9.73 | 1/ 60 | 148 | 72.53 | 0/ 60 | 2217 | 2061.56 | 1/ 22 | 47277 |
| MLP | 9.98 | 0/ 60 | 157 | 77.48 | 0/ 60 | 2299 | 1984.26 | 1/ 24 | 40188 |

(b) Set Cover

# 10 Scaling to 2 × Big instances

In this section we investigate the generalization power of FiLM models trained on dataset obtained from small instances. Specifically, we generate 20 random instances of size double that of big instances, and use the trained models of FiLM and GNN to compare their performance against RPB. We use the time limit of 7200 seconds, i.e. 2 hours to account for longer solving running times as one scales out to bigger instances. We observe that the power to generalize is largely dependent on the problem family. For example, FiLM models can still outperform other strategies on scaling out on capacitated facility location problems. However, we found that RPB remains competitive on setcover problems. We note that this shortcoming of the FiLM models can be overcome via larger size of hidden layers and training on slightly larger instances than what has been used in the main paper. Table 9 shows these results.

Table 9: Performance of branching strategies on twice the size of biggest instances (2 × Big) considered in the main paper. 20 "Bigger" instances were solved using 3 seeds each resulting in a total of 60 runs. We see that FiLM models still remain competitive, and it is highly dependent on the family of problem.

| | facilities | | | setcover | | |
| Model | Time | Wins | Nodes | Time | Wins | Nodes |
|---|---|---|---|---|---|---|
| RPB | 7200.14 | 0 / 0 | n/a | **6346.17** | **9** / 12 | 135 132 |
| GNN | 7111.83 | 1 / 5 | n/a | 7200.25 | 0 / 0 | n/a |
| FiLM (ours) | **7052.27** | **4** / 4 | n/a | 6508.78 | 3 / 10 | **107 187** |
| GNN-GPU | 6625.55 | – / 13 | n/a | 6008.14 | – / 12 | 93 909 |

# 11 Effect of different training protocols on B&B performance

Table 10 shows the performance of different training protocols on B&B performance. Although the models trained with auxiliary tasks (AT) and knowledge distillation (KD) are a clear winner up until problem sets of size medium, there is a tie between the models trained with KD and those trained with KD & AT. However, it is worth noting that the difference in B&B performance across different training protocols is not huge, and such performance evaluation can be read from the test accuracy of

Table 10: Effect of training protocols on the performance of *branching strategies*. We report geometric mean of solving times, number of times a method won (in solving time) over total finished runs, and geometric mean of number of nodes. Refer to section 5 (main) for more details. The best performing results are in **bold**. *Models were regularized to prevent overfitting on small instances.

| Model | Small Time | Small Wins | Small Nodes | Medium Time | Medium Wins | Medium Nodes | Big Time | Big Wins | Big Nodes |
|---|---|---|---|---|---|---|---|---|---|
| e2e | 25.05 | 22 / 60 | 114 | 154.12 | 4 / 60 | 340 | 550.03 | 15 / 57 | 339 |
| e2e + KD | 28.00 | 2 / 60 | 111 | 143.12 | 18 / 60 | 343 | **507.50** | **26** / 57 | **325** |
| e2e + KD + AT | **24.67** | **36** / 60 | **109** | **136.42** | **38** / 60 | 336 | 531.70 | 16 / 57 | 345 |
| Capacitated Facility Location | | | | | | | | | |
| e2e | 9.70 | 1 / 60 | 152 | 71.18 | 1 / 60 | 2186 | 1869.57 | 4 / 25 | **40 341** |
| e2e + KD | 9.81 | 0 / 60 | **146** | 70.88 | 4 / 60 | 2173 | **1842.29** | **15** / 25 | 40 437 |
| e2e + KD + AT | **8.73** | **59** / 60 | 147 | **63.75** | **55** / 60 | **2169** | 1843.24 | 8 / 26 | 40 881 |
| Set Covering | | | | | | | | | |
| e2e | 2.45 | 1 / 60 | **72** | 17.59 | 5 / 60 | 702 | 225.88 | 17 / 60 | 8939 |
| e2e + KD | 2.35 | 0 / 60 | 73 | 17.59 | 2 / 60 | 720 | 241.95 | 7 / 59 | 8846 |
| e2e + KD + AT | **2.13** | **59** / 60 | 73 | **15.71** | **53** / 60 | **686** | **217.02** | **36** / 60 | **8711** |
| Combinatorial Auctions | | | | | | | | | |
| e2e | 205.63 | 2 / 54 | 559 | 1103.47 | 1 / 29 | 1137 | 2457.94 | 1 / 4 | **2268** |
| e2e + KD | 333.52 | 1 / 52 | 486 | 926.12 | 1 / 41 | 604 | 2503.65 | 0 / 7 | 1953 |
| e2e + KD + AT | **52.96** | **54** / 55 | **410** | **131.45** | **54** / 54 | **331** | **1823.29** | **14** / 15 | 3049 |
| Maximum Independent Set* | | | | | | | | | |

trained models. For example, as evident in Table 5, difference in test accuracy of FiLM (e2e & KD) and FiLM (e2e & KD & AT) is not significant for problem sets - Capacitated Facility Location and Set Covering, but its significant enough for problem sets - Combinatorial Auctions and Maximum Independent Set. **Given that the inference cost is independent of training protocols, to attain better generalization performance, we recommend FiLM (e2e & KD & AT) only when these models have significantly better accuracy than FiLM (e2e & KD).**