[Reviews · NeurIPS 2020]

Review 1

Summary and Contributions: This paper proposes an efficient deep learning model performing branching in Branch&Bound, which is runnable on CPU and comparable to other GPU heavy approach. In order to achieve this efficiency, the author paid attention to the expensive inference of the successful existing work using GNN and propose a model that combines GNN with MLP.

Strengths: - The proposed hybrid model only using CPU is able to achieve comparable results to the results from full GNN using GPUs, which makes this ML-based MILP method accessible to MILP practitioners. - The tricks that were tried to improve generalization are explained in detail and the limitation and capability of the proposed method are well-discussed. ============================================================ ============================================================ I have read the response and been involved in the dicussion. Thanks for your response. I appreciate its practical value in spite of seeming a bit weak novelty. I maintain my score.

Weaknesses: -

Correctness: - The paper is mostly empirical and experimental details are provided properly. - Which data are used in Figure 3 and Table 2? Small in Table 4 or different one than ones used in Table 4?

Clarity: - The paper is well-written which make it easy to follow. - Typos 313 : 'adressed' -> 'addressed' 313 : 'cmall' -> 'small'

Relation to Prior Work: This work is on ML method for branching deployable on CPU-only environment. Even though similar hybrid approaches have been taken in other applications, this work focuses on MILP with scalability. These are well-addressed in 'Related Work'

Reproducibility: Yes

Additional Feedback:


Review 2

Summary and Contributions: The paper proposes a hybrid model for branch variable selection in mixed-integer programming. The hybrid model uses a GNN from prior work to extract features from the root and combines them with node-specific features via a computationally inexpensive MLP into a prediction. The main contribution is experimental, as the authors test variants of their model and different training strategies on instances from different problem types.

Strengths: The work is practically relevant, as in mixed-integer programming an important trade-off has to be found between the time it takes to determine a branch variable and its effectiveness. The authors show that the proposed hybrid model is favorable when only a CPU is available for the inference.

Weaknesses: While the authors show some improvements in computation time, I am a bit doubtful of the significance of these results. The main points of my critique are: 1. The performance improvements are mainly due to faster inference (according to Table 4). 2. The model is a combination of prior approaches, hence the novelty is arguable. 3. The additional experimental results regarding the training protocol and the loss weighting are not really convincing (Table 2 and Table 3). In line 260, the authors claim that the rather insignificant accuracy improvement in Table 2 translates to "substantial" reduction in the number of B&B nodes. On which numbers is this claim based? I did not see support for this. The differences in Table 3 are rather small as well, so I am skeptical about the insight gained here. All in all, this paper constitutes good experimental work, but I am not sure the results justify acceptance. I am happy to be convinced otherwise. --- Update --- I have considered the author reply and the discussion. At large, my view of the paper has not changed much. I think, for the scientific community, the insight provided by this work is relatively small. It does not exactly advance the state-of-the-art model wise (ie. in prediction quality) or running time wise at the same prediction quality. Instead, it combines prior approaches (albeit it in a novel way) in order to trade off model quality with inference time in a restricted hardware setting. The unavailability of performance GPU cards to practitioners is not a strong argument for significance, in my opinion. I think it is unlikely that those practitioners who are willing to invest in the application of the state-of-the-art in this research area are unwilling to invest in the necessary hardware. Given the pace at which the field and the underlying hardware advances, I am afraid the half life of this contribution might be rather short. There may be other technical aspects to consider concerning the integration of CPU-heavy and GPU-heavy computations. I do not know how important this really is in practice and the authors have not addressed it sufficiently to convince me. All in all, I think this paper is a solid piece of technical work, but I remain a bit doubtful that publication at NeurIPS is merited. Since the other reviewers are somewhat more positive I decided to raise my score slightly to 5 in order to move towards consensus, but I am not arguing for acceptance.

Correctness: The methodology appears to be correct.

Clarity: The paper is generally a good read.

Relation to Prior Work: Prior work is discussed appropriately.

Reproducibility: Yes

Additional Feedback: The authors write that the hybrid SVM-based models outperform the MLP in terms of accuracy (line 241 - 257). Considering Figure 3, I respectfully disagree that this conclusion can be reached. I suggest to rephrase line 255: "However, there is no clear winner among them".


Review 3

Summary and Contributions: The paper suggests a hybrid imitation learning approach between GNNs and MLPs for optimizing the branching scheme of the branch and bound algorithm. The paper shows empirically that this results in better branching strategies with fewer number of expanded nodes and improvement in runtime.

Strengths: The advantages of hybrid models from the computational side are clear, using high-level features from the root node from a heavier model and a simpler model within the tree makes sense, and this is confirmed empirically on different combinatorial problems and different variants of hybrid models. All of the small tweaks to the method (loss function weighting dependent on depth, end-to-end training of root model, knowledge distillation and auxiliary tasks) are mentioned and analyzed in an ablation study and are clearly motivated in the main text of the paper. The runtime improvements are considerable.

Weaknesses: I don’t have many complaints about the paper, it is overall well-rounded. Just that the small tweaks introduced seem to not have a big impact on accuracy as per Table 2, it seems to be problem dependent and oftentimes the performance improvement is within the deviation. The performance of hybrid approaches relies heavily on feature extraction at the root node, concretely on the approach proposed in [17]. There should be more discussion on the method used for pre-training and the trafeoff between accuracy and model complexity.

Correctness: The claims are backed by experimental results and all the relevant metrics for this type of problem have been evaluated (runtime, number of expanded nodes, number of wins). Table 2 needs to be further commented, it seems as if the main benefit comes from AT(auxiliary tasks) and there is no experiment demonstrating the improvement resulting from only using AT.

Clarity: The paper is well-written, the method is clear. There are just some details missing, such as the pre-training procedure of the GNN (even if it were included in the supplementary) and the architecture of the GNN. Why does knowledge distillation benefit generalization? This is not really clear from the results in Table 2, in 2/4 problems introducing KD actually hurts the test accuracy.

Relation to Prior Work: To the extent of my knowledge, the related work has been covered enough and the work that motivated the current paper is connected.

Reproducibility: No

Additional Feedback: 12: as far as I know, the term MLP refers to multi-layer perceptron and not multi-linear? 212: typo “sigmodal” -> “sigmoidal” 313: typo “cmall” -> “small” 316: weird formulation “larger instances than (small)” Table 2: I think that there should be an experiment with AT only, to show how much performance is attributed to it. Don’t forget to change the abstract in the camera-ready, 26% vs reduction of 26% is a big difference! There is a need for exploring the model complexity vs. test performance tradeoff that is not covered enough in the paper, especially the comparison between root model complexity and inner model complexity. Please describe the model architectures better (GNN and MLP), this can be done in the supplementary material. Perhaps an additional evaluation metric to consider is the optimality gap after time limit? The test accuracy in the B&B node expansion might not reflect the actual performance that we are looking for. --update-- Since the other reviewers turned my attention towards the lack of novelty, and I do not consider myself an expert in this area, I am lowering my score a bit. But I still think that it is an ok submission.


Review 4

Summary and Contributions: The paper proposes a hybrid architecture that uses a GNN model at the node node of the B&B tree and a week but fast MLP model at the leaf nodes. The authors studied the impact of a few variants to their training protocol for learning to branching. The variants include end-to-end training, knowledge distillation, auxiliary tasks, and depth-dependent weighting of the training lost. A number of experiment results are presented in the paper. The paper reports results of four types of problem: capacitated facility location, combinatorial auction, set covering, and independent set. The results show the hybrid method is much less expensive than GNN model, and achieves 26% time reduction comparing to the default branching method in SCIP.

Strengths: The topic is interesting and important in numerical optimization research. The proposed hybrid method is new and less expensive than GNN and seems still effective in reducing solution time. The authors studies the impact of knowledge distillation, auxiliary tasks, and loss wighting. The results presented in the paper is encouraging, although the problems are randomly generated. It will be certainly interesting to test its performance on real world instances.

Weaknesses: The method proposed is specific to problem types. Different problem types require different models. It is not a general branching strategy. How to generalize this method for generic MILPs is not an easy task. SCIP's pseudo-cost branching, etc. are general methods and are trying to do simple online learning. First, I think it maybe not too fair to compare special methods with general methods . Second, online learning can be expensive and but can adapt to specific instance situation better. Even a simple version like reliability branching seems to be very effective. If we also tune the hyper parameters in reliability branching, I guess we can also get better results for specific problem types.

Correctness: yes. looks correct to me.

Clarity: The paper is written well.

Relation to Prior Work: The paper reviews existing literature, explains the difference with other researches.

Reproducibility: No

Additional Feedback: ==After Rebuttal== Thanks authors for the feedback and answer to my questions. I'm fine with most points, but I still think pseudo-cost branching is on-line learning. Maybe I'm wrong, but here is what my understanding. The pseudo-cost value of a decision variable is updated whenever the variable is branched on. The pseudo-costs of decision variables are like the parameters of a ML model, so the pseudo-cost branching model is evolving during the solution process.

[Author Response · NeurIPS 2020]

We thank reviewers for their time and valuable comments. We will try to answer the concerns raised by each reviewer.

We thank R1 for their positive feedback. F3 and T2 are indeed results on *small* instances. We will clarify the captions.

We thank R2 for their valuable comments. **1)** We respectfully disagree on 1., as we do not consider this as a weakness of the paper. As we explained in l.52-60, state-of-the-art MILP solvers are CPU-based, and performance of these solvers is benchmarked in terms of solving time. As such, fast inference is crucial for any ML-based branching strategy to be competitive w.r.t. this metric. Devising an effective model that can learn how to branch accurately is one thing, and making such a model practical, in realistic scenarios where practitioners do not have access to high-end GPU cards, is another. The high inference cost of GNNs was

Table A1: Effect of training protocols on B&B tree size for the FiLM model (geometric mean over commonly solved instances).

|  | cauctions | | facilities | |
|  | medium | big | medium | big |
| --- | --- | --- | --- | --- |
| e2e | 702 | 8939 | 340 | 339 |
| e2e & KD | 720 | 8846 | 343 | **325** |
| e2e & KD & AT | **686** | **8711** | **336** | 345 |

one of the main limitations for integration within MILP solvers, which we believe we addressed in the submission. Our proposed hybrid model offers a better computational trade-off, while retaining most of the accuracy and generalization ability of GNNs. We propose to make a better case of this question, which we will discuss more thoroughly in a new sub-section "The accuracy/complexity trade-off". **2)** We also respectfully disagree on 2. Taken individually, it is true that each part of our proposed methodology has been proposed in a different ML context. However the core idea of running an expressive, but expensive, model once at the root node, and using the resulting transformation to modulate a simpler model everywhere else in the tree, is novel. We believe it is a natural idea, specific to the problem of deploying ML models in time-sensitive B&B environments, which the ML/OR community will find interesting. The paper shows that this intuitive idea is practical, and promising for the field. **3)** We agree on 3., as our training protocol improvements have only a minor impact on the final performance of the model for branching. We conducted additional preliminary experiments (Table A1), and it seems like KD has an effect on generalization which is visible only on hard instances, while AT has a globally positive effect (except for one scenario in Table A1). We propose to lower our claim from "substantial" to "minor", and we will conduct and report the complete experiment in the final version of the paper, including the small instances and the other 2 problem families. However, we would like to point out that the training protocol improvements do not constitute the main contribution of the paper. **4)** Finally, we thank R2 for the suggestion regarding l.255, which more accurately reflects the behaviour of hybrid-SVM models.

We thank R3 for their constructive comments. **1)** We agree that our training protocol tweaks do have a minor impact, and we will make this more clear in the text (see response 3 to R2). **2)** Regarding the trade-off between model accuracy and complexity, we agree that our discussion in l.52-60 is too brief, and will put more emphasis on this discussion in a new sub-section (see answer 1 to R2). **3)** Regarding KD, we will report additional experiments to better highlight its contribution (see answer 3 to R2). **4)** We agree that a separate experiment with only AT would bring value to the paper. We will conduct it and report the results in the supplementary materials, or the main paper if space permits. **5)** We will fix the two abstract misphrasings, thank you. **6)** We did measure the suggested additional evaluation metric, optimality gap at time out (see Table A2). The results are again in favor of our proposed approach, we will include them in the paper. Thank you for the suggestion.

Table A2: Mean optimality gap (lower the better) of commonly unsolved "big" instances (number of such instances brackets).

|  | setcover (33) | indset (39) |
| --- | --- | --- |
| FSB | 0.170944 | 0.075543 |
| PB | 0.071286 | 0.029791 |
| RPB | 0.062841 | 0.025186 |
| COMP | 0.074041 | 0.025214 |
| GNN | 0.103906 | 0.034104 |
| FiLM | **0.059692** | **0.018685** |

We thank R4 for their insightful comments. **1)** The proposed method is indeed specifically trained on one problem type, which is a very valid point that we do not discuss in the paper. We follow the same setting as Gasse et al., who consider this scenario as relevant because most practitioners usually only care about solving very specific problem types. We will add a reference to that argument to clarify this point. **2)** Regarding "different problem types require different models", this might be true, but here we provide evidence that goes in the other direction. Indeed, the same model seems to work reasonably well on a variety of problem types, without the need for specific adaptations (except for re-training). **3)** We agree that our model does not yield a general MILP branching strategy. Attaining this goal with ML would be a substantial achievement, and indeed is not an easy task. We will include this remark in the concluding section. **4)** Our comparison to general MILP branching strategies like RPB is not completely fair. It is a very good point. However, developing specialized versions of such strategies is a challenge in itself, and would require a dedicated paper. We would gladly compare to such approaches, if available. Thank you for raising this point, and we will include this discussion as well. **5)** Regarding online learning, we do not share the same view here. Those branching rules exploit non-local LP information to improve future decisions based on past branching decisions. But our model exploits such information as well, as pseudo-cost is also used as an input feature, at every node. Online learning would imply changing the branching strategy behaviour given the exact same features, which, in our knowledge, none of the branching strategies do.

Finally, both R3 and R4 do not consider our work reproducible, despite the source code being provided. We take note, and will provide more exhaustive implementation details in the appendix of the final paper.

[Meta-Review · NeurIPS 2020]

The paper proposes a CPU version based on a recent machine learning algorithm for selecting branching variables in branch and bound for ILPs. The hybrid method is based on a GNN together with a neural net to form a prediction. The reviewers agree on the practical benefits this brings and agree on the work being a solid experimental and engineering work. Reviewers criticize that the work is incremental and the ideas used are found in previous work. One reviewer questions the practical significance of the work for the future. The overall recommendation for the paper is accept, since the paper overall represents an interesting practical contribution,